# Bayesian Distributed Stochastic Gradient Descent

**Michael Teng**
Department of Engineering Sciences
University of Oxford
mteng@robots.ox.ac.uk

**Frank Wood**
Department of Computer Science
University of British Columbia
fwood@cs.ubc.ca

## Abstract

We introduce Bayesian distributed stochastic gradient descent (BDSGD), a high-throughput algorithm for training deep neural networks on parallel computing clusters. This algorithm uses amortized inference in a deep generative model to perform joint posterior predictive inference of mini-batch gradient computation times in a compute cluster specific manner. Specifically, our algorithm mitigates the straggler effect in synchronous, gradient-based optimization by choosing an optimal cutoff beyond which mini-batch gradient messages from slow workers are ignored. The principle novel contribution and finding of this work goes beyond this by demonstrating that using the predicted run-times from a generative model of cluster worker performance improves over the static-cutoff prior art, leading to higher gradient computation throughput on large compute clusters. In our experiments we show that eagerly discarding the mini-batch gradient computations of stragglers not only increases throughput but sometimes also increases the overall rate of convergence as a function of wall-clock time by virtue of eliminating idleness.

## 1 Introduction

Deep learning success stories are predicated on large neural network models being trained using ever larger amounts of data. While the computational speed and memory available on individual computers and GPUs continually grows, there will always be some problems and settings in which the amount of training data available will not fit entirely into the memory of one computer. What is more, and even for a fixed amount of data, as the number of parameters in a neural network or the complexity of the computation it performs increases, so too do the incurred economic and time costs to train. Both large training datasets and complex networks inspire parallel training algorithms.

In this work we focus on parallel stochastic gradient descent (SGD). Like the substantial and growing body of work on this topic (Recht et al. (2011); Dean et al. (2012); McMahan and Streeter (2014); Zhang et al. (2015)) we too focus on gradient computations computed in parallel on "mini-batches" drawn from the training data. However, unlike most of these methods which are *asynchronous* in nature, we focus instead on improving the performance of *synchronous* distributed SGD, very much like Chen et al. (2016), upon whose work we directly build.

A problem in fully synchronous distributed SGD is the *straggler effect*. This real-world effect is caused by the small and constantly varying subset of worker nodes that, for factors outside our control, perform their mini-batch gradient computation slower than the rest of the concurrent workers, causing long idle times in workers which already have finished. Chen et al. (2016) introduce a method of mitigating the straggler effect on wall-clock convergence rate by picking a fixed cut-off for the number of workers on which to wait before synchronously updating the parameter on a centralized parameter server. They found, as we demonstrate in this work as well, that the increased gradient computation throughput that comes from reducing idle time more than offsets the loss of a small fraction of mini-batch gradient contributions per gradient descent step.

Our work exploits this same key idea but improves the way the likely number of stragglers is identified. In particular we instrument and generate training data once for a particular compute cluster and neural network architecture, and then use this data to train a lagged generative latent-variable time-series model that is used to predict the joint run-time behavior of all the workers in the cluster. For highly contentious clusters with poor job schedulers, such a model might reasonably be expected to learn to model latent states that produce correlated, grouped increases in observed run-times due to resource contention. For well-engineered clusters, such a model might learn that worker run-times are nearly perfectly independently and identically distributed.

Specifying such a flexible model by hand would be difficult. Also, we will need to perform real-time posterior predictive inference in said model at runtime to dynamically predict straggler cutoff. For both these reasons we use the variational autoencoder loss (Kingma and Welling, 2013) to simultaneously learn not only the model parameters but also the parameters of an amortized inference neural network (Ritchie et al., 2016; Le et al., 2017) that allows for real-time approximate predictive inference of worker run-times.

The main contributions of this paper are:

- The idea of using amortized inference in a deep state space model to predict compute cluster worker run-times, in particular for use in a distributed synchronous gradient descent algorithm.
- The BDSGD algorithm itself, including the approximations made to enable real-time posterior predictive inference.
- The empirical verification at scale of the increased gradient computation throughput that our algorithm yields when training deep neural networks in parallel on large clusters.

The rest of the paper is organized as follows. In section 2, we give necessary background on why and how synchronous distributed SGD can be improved. In section 3, we explain our choice of generative model for cutoff determination. In section 4, we present our experimental results.

## 2 Background and Motivation

In stochastic gradient descent, we use unbiased estimates of gradients to update parameter settings. Synchronous distributed SGD differs from single-threaded mini-batch SGD in that the mini-batch of size $m$ is distributed to $N$ total workers that locally compute sub-mini-batch gradients before communicating the result back to a centralized parameter server that updates the parameter vector using an update rule:

$$\theta^{(t+1)} = \theta^{(t)} - \alpha \frac{1}{N} \sum_{i=1}^{N} f\left(\theta^{(t)}, (i-1)\frac{m}{N}, i\frac{m}{N}\right) \tag{1}$$

with

$$f(\theta, a, b) = \frac{1}{b-a} \sum_{k=0}^{b-a} \nabla_{\theta^{(t)}} F(\theta, z^{(k)}, y^{(k)})$$

where $\theta$ are the network parameters, $F$ is the loss function, and $\alpha$ is the learning rate. Although not shown, asynchronous SGD is lock-free, and parameter updates are made whenever any worker reports a sub-mini-batch gradient to the parameter server which results in out-of-date or stale gradient information (Recht et al., 2011). Unlike asynchronous distributed SGD, synchronous distributed SGD is equivalent to single-threaded SGD with batchsize, $m$. This allows hyperparameters, $\alpha$ and $m$, to be tuned in the distributed setting without having to consider the possibility of stale gradients (Hoffer et al., 2017).

### 2.1 Effect of Stragglers

In synchronous SGD, we can attribute low throughput, in the sense of central parameter updates per unit time, to the straggler effect that arises in real-world cluster computing scenarios with multiple workers computing in parallel. Consider Equation 1, in which $f(\theta, \cdot, \cdot)$ is computed independently on an memory-isolated logical processor. Let $x_j$ be the time it takes for $f$ to be computed on the

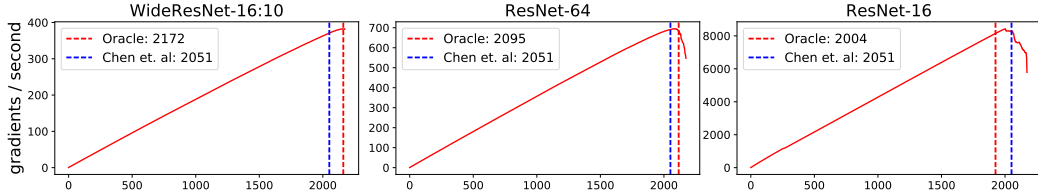

Figure 1: Oracle throughput curves (best achievable in hindsight) for synchronous SGD runs for three different neural networks on the same 2175-worker cluster. From left: low variance (1-2% stragglers), medium variance (2-4% stragglers), and high variance (8-12% stragglers) throughput curves with mean±std worker gradient computation times being $5.34 \pm 0.13$ seconds, $2.83 \pm 0.077$ seconds, and $0.24 \pm 0.018$ seconds, respectively. The x-axes are the number of workers. The y-axes show throughput achieved if all workers beyond the x-axis value are ignored. When runtimes for gradient computations have low variance relative to total runtime, Chen et al. underestimates the optimal cutoff point whereas when runtimes have proportionally higher variance, Chen et al. over-estimates the optimal cutoff point. Our approach achieves more accurate estimates of the optimal cutoff in both scenarios.

worker indexed by $j$ for $j \in 1...N$. Distributed compute clusters are not perfect, otherwise $x_j$ would be a constant, independent of $j$, and all workers would finish at the same time and only be idle while the parameter server aggregates the gradients and sends back the new parameters. Instead $x_j$ is actually random. Moreover, the joint distribution of all the $x_j$'s is likely, again in real-world settings, to be non-trivially correlated owing to cluster architecture and resource contention. For instance, most modern clusters consist of computers or graphics processing units each in turn having a small number of independent processors, so slow-downs in one logical processing unit are likely to be exhibited by others sharing the same bus or network address. What is more, in modern operating systems, time-correlated contention is quite common, particularly in clusters under queue management systems, when, for instance, other processes are concurrently executed. All this yields worker compute times that may be non-trivially correlated in both time and in "space".

Our aim is to significantly reduce the effect of stragglers on throughput and to do so by modeling cluster worker compute times in a way that intelligently and adaptively responds to the kinds of correlated run-time variations actually observed in the real world. What we find is that doing so improves overall performance of distributed mini-batch SGD.

## 3 Methodology

Our approach works by maximising the total throughput of parameter updates during a distributed mini-batch SGD run. The basic idea, shared with Chen et al. (2016), is to predict a cutoff, $c_t < N$, for each iteration of SGD which dictates the total number of workers on which to wait before taking a gradient step in the parameter space. While Chen et al. (2016) use a fixed cutoff, $c_t = 0.943 \cdot N$; $\forall t$, we would like for $c_t$ to be evolving dynamically with each iteration and in a manner specific to each compute cluster, neural network architecture pair. We note that for overall rate of convergence, throughput is not the exact quantity we wish to maximize; that being some quantity related to the rate of expected gain in objective function value instead, but it is the proxy we use in this work. Also, paradoxically, lower throughput, by virtue of smaller mini-batch sizes, may in some instances increase the rate of convergence, an effect previously documented in the literature (Masters and Luschi, 2018).

The central considerations are: what is the notion of throughput we should optimize? And how do we predict the cutoff that achieves it? Simply optimizing overall run-time admits a trivial and unhelpful solution of setting $c_t = 0$. Each iteration and the overall algorithm would then take no time but achieve nothing. Instead we seek to maximize the number of workers to finish in a given amount of time, i.e. throughput $\Omega(c)$, which we define to be:

$$\Omega(c) = \frac{c}{\tilde{x}_{(c)}}$$

where $c$ indexes the *ordered* worker run-times $\tilde{x}_{(c)}$. Note that, for now, we avoid indexing run-times by SGD loop iteration. Soon, we will address temporal correlation between worker runtimes.

With this definition, we can plot throughput curves that show how throughput drops off as the straggler effect increases (Figure 1). On the well-configured cluster used to produce Figure 1, a high percentage of workers (between 80-95%) finish at roughly the same time, so in this regime, throughput of the system increases linearly for each additional worker. However, continuing to wait for more workers includes some stragglers which eventually decreases the overall throughput.

We define our objective to be to maximize the throughput of the system as defined above, i.e. $\arg\max_c \Omega(c)$ at all times. This also implicitly handles the tradeoff between iteration speedup and the learning signal reduction that comes from using a higher variance gradient estimate given by discarding gradient information.

Setting the cutoff optimally and dynamically requires a model which is able to learn and predict the joint ordered run-times of all cluster workers. With such a model, we can make informed and accurate predictions about the next set of run-times per worker and consequently make a real-time, near-optimal choice of $c$ for the subsequent loop of mini-batch gradient calculations. How we model compute cluster worker performance follows.

## 3.1 Modeling Compute Cluster Worker Performance

As before, let $x_j \in \mathbb{R}^+$ be the time it takes for $f$ to be computed on the worker indexed by $j$. Assume that these are distributed according to some distribution $p$. Given a set of $n$, $p(\cdot)$-distributed random variables $x_1, x_2, \ldots, x_n$ we wish to know the joint distribution of the $n$ *sorted* random variables $\tilde{x}_{(1)}, \tilde{x}_{(2)}, \ldots, \tilde{x}_{(n)}$. Such quantities are known as "order statistics." Each $p(\tilde{x}_{(j)})$ describes the distribution of the $j^{th}$ largest sorted run-time under independent draws from this underlying distribution. Taking the mean of each order statistic allows us to derive a cutoff using our notion of throughput, given as:

$$\arg\max_c \Omega(c) = \arg\max_c \mathbb{E}\left[\frac{c}{\tilde{x}_{(c)}}\right] \tag{2}$$

### 3.1.1 Elfving Cutoff

The first model of runtimes we consider assumes that they are are independent and identically distributed (iid) Gaussian. Under the assumption that $x_j = \mathcal{N}(\mu_x, \sigma_x^2)$ the distribution of each order statistic $p(\tilde{x}_{(1)}), p(\tilde{x}_{(2)}), \ldots, p(\tilde{x}_{(n)})$ is independent and $\mathbb{E}[\tilde{x}_{(1)}] \leq \mathbb{E}[\tilde{x}_{(2)}], \ldots, \leq \mathbb{E}[\tilde{x}_{(n)}]$.

Under the given iid normality assumption the distribution of the each order statistic has closed form:

$$p(\tilde{x}_{(j)}) = Z(n, j) \int_{-\infty}^{\infty} x [\Phi(x)]^{j-1} [1 - \Phi(x)]^{n-j} p(x) dx$$

where $\Phi(x)$ is the cumulative distribution function (CDF) of $\mathcal{N}(\mu_t, \sigma_t^2)$ and $Z(n, j) = \frac{n!}{(j-1)!(n-j)!}$. Note that each order statistic's distribution, including the maximum, increases as the variance of the run-time distribution increases, while the average run-time does not.

As a baseline in subsequent sections we will use an approximation of the expected order statistics under this iid normality assumption. This is known as the Elfving (1947) formula (Royston, 1982):

$$\mathbb{E}[\tilde{x}_{(j)}] \approx \mu_t + \Phi^{-1}\left(\frac{n - \frac{\pi}{8}}{j - \frac{\pi}{4} + 1}; 0, 1\right)\sigma_t \tag{3}$$

Here, we note that the Elfving model requires full observability of runtimes to predict subsequent runtimes in a production setting. In practice, the parameters $\mu_t, \sigma_t$ in Eqn. 3 are fit using maximum likelihood on the first fixed lagged window and remain static during the remainder of the run.

While some clusters may approximate the strong assumptions required to use the Elfving formula for cutoff prediction, most compute clusters will emit joint order statistics of non-Gaussian distributed correlated random variables, for which no analytic expression exists. However, if we have a predictive model of the joint distribution of the $x_j$'s (or $\tilde{x}_{(j)}$'s), we can use sorted samples from such a joint distribution to obtain a Monte Carlo approximation of the order-statistics. In the next section, we will detail how to construct the predictive model in order to learn correlations of worker runtimes.

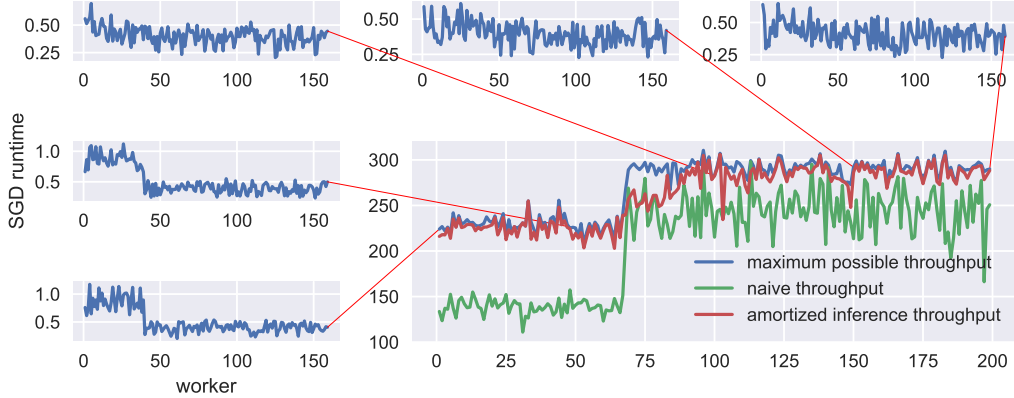

Figure 2: Predicted throughputs. Each runtime plot (5 surrounding the top figure) shows the individual runtimes of all workers (x-axis index) during an iteration of SGD on a 158 node cluster. We highlight SGD iterations 1, 50, 100, 150, and 200 which highlight two significantly different regimes of persistent time-and-machine-identity correlated worker runtimes. The bottom-right figure displays a comparison of throughputs achieved at each of the 200 SGD iterations by waiting for all workers to finish (green) and using our approach, BDSGD (red), relative to the ground truth maximum achievable (blue). BDSGD predicts cutoffs that achieve near optimal throughput, in a setting where fixed-cutoffs are insufficient and Elfving assumptions do not hold.

### 3.1.2 Bayesian Cutoff

In this section, we formally introduce our proposed training method, which we call Bayesian distributed SGD (BDSGD). Before introducing the design of the generative model we use to predict worker run-times, first consider the practical implications of using a generative model instead of a purely autoregressive model. In short we can only consider worker run-time prediction models that are extremely sample efficient to train. We also can only consider a kind of model that allows real-time prediction because it will be in the inner loop of the parameter server and used to predict at run-time how many straggling workers to ignore. Deep neural net auto-regressors satisfy the latter but not the former. Generative models satisfy the former but historically not the latter; except now deep neural net guided amortized inference in generative models does. This forms the core of our technical approach.

We will model the time sequence of observed joint worker run-times $\boldsymbol{x}_{T-\ell}, \ldots, \boldsymbol{x}_T$ using a deep state space model where $\boldsymbol{z}_{T-\ell}, \ldots, \boldsymbol{z}_T$ is the time evolving unobserved latent state of the cluster. In this framework, $\boldsymbol{x}_{T-\ell:T}$ may be replaced with directly modeling $\tilde{\boldsymbol{x}}_{T-\ell:T}$, and we continue with $\boldsymbol{x}_{T-\ell:T}$ for clarity. The dependency structure of our model factorizes as

$$p_\theta(\boldsymbol{x}_{T-\ell:T}, \boldsymbol{z}_{T-\ell:T}) = \prod_{i=T-\ell}^{T} p_\theta(\boldsymbol{z}_i|\boldsymbol{z}_{i-1}) \prod_{i=T-\ell}^{T} p_\theta(\boldsymbol{x}_i|\boldsymbol{z}_i)$$

where, for reasons specific to amortizing inference, we will restrict our model to a fixed-lag $\ell$ window. The principle model use is the accurate prediction of the next set of worker run-times from those that have come before:

$$p(\boldsymbol{x}_{T+1}|\boldsymbol{x}_{T-\ell:T}) = \int p_\theta(\boldsymbol{x}_{T+1}|\boldsymbol{z}_{T+1})p_\theta(\boldsymbol{z}_{T+1}|\boldsymbol{z}_T)p(\boldsymbol{z}_{T-\ell:T}|\boldsymbol{x}_{T-\ell:T})d\boldsymbol{z}_{T-\ell:T+1} \quad (4)$$

### 3.1.3 Model Learning and Amortized Inference

With the course-grained model dependency defined, it remains to specify the fine-grained parameterization of the generative model, to explain how to train the model, and to show how to perform real-time approximate inference in the model.

We use the deep linear dynamical model introduced by Krishnan et al. (2017), that constructs the LDS with MLP link functions between Gaussian distributed latent state and observation vectors. Inference in the model is done with a non-Markovian proposal network. Namely, the transition and

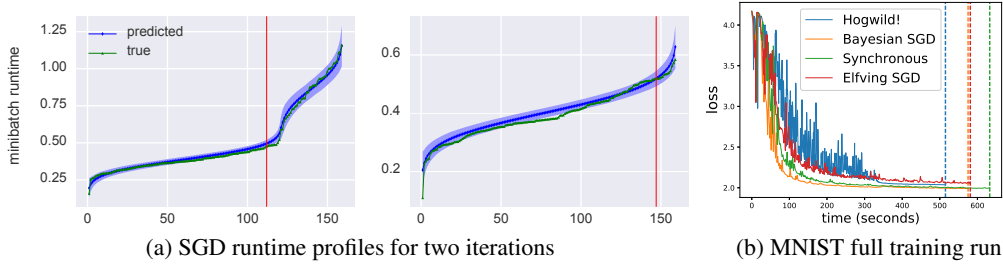

(a) SGD runtime profiles for two iterations  (b) MNIST full training run

Figure 3: Cumulative results of 158-worker cluster on 3-layer MLP training. The left two figures are plots of observed runtimes vs predicted runtime order statistics of two iterations of SGD of the validation set in the BDSGD training step. The maximum throughput cutoff under the model predictions is shown in red, indicating a large chunk of idle time is reduced as a result of stopping early. Notably, when there are exceptionally slow workers present, the cutoff is set to proceed without any of them as seen in left figure of subplot (a). Subplot (b) shows MNIST validation loss for model-based methods, Elfving and BDSGD, compared to naive synchronous (waiting for all workers) and asynchronous (Hogwild) approaches, where dashed vertical lines indicate the time at which the final iteration completed (all training methods perform the same number of mini-batch gradient updates). In the lower right corner of subplot (b), we observe that BDSGD achieves the fastest time to complete the fixed number of gradient updates of the synchronous methods, while also achieving the lowest validation loss.

emission functions in our model are parameterized by neural networks, the inference over which is guided by an RNN. For a detailed exposition, see the supplementary materials.

The flexibility of such a model allows us to avoid making restrictive or inappropriate assumptions that might be quite far from the true generative model while imposing rough structural assumptions that seem appropriate like correlation over time and correlation between workers at a given time.

The remaining tasks are to, given a set of training data, i.e. fully observed SGD runtimes specific to a cluster, learn $\theta$ *and* train an amortized inference network to perform realtime inference in said model. For this we utilize the variational autoencoder-style loss used for amortized inference in deep probabilistic programming with guide programs (Ritchie et al., 2016).

We use stochastic gradient descent to simultaneously optimize the variational evidence lower bound (ELBO) with respect to both $\phi$ and $\theta$:

$$\text{ELBO} = \mathbb{E}_{q_\phi(\boldsymbol{z}_{T-\ell:t}|\boldsymbol{x}_{T-\ell:T})} \log \left( \frac{p_\theta(\boldsymbol{x}_{T-\ell:t}, \boldsymbol{z}_{T-\ell:t})}{q_\phi(\boldsymbol{z}_{T-\ell:t}|\boldsymbol{x}_{T-\ell:T})} \right)$$

where

$$q_\phi(\boldsymbol{z}_{T-\ell:t}|\boldsymbol{x}_{T-\ell:T}) = \prod_{t=T-\ell}^{T} q_\phi(\boldsymbol{z}_t|\boldsymbol{z}_{T-\ell:t-1}, \boldsymbol{x}_{T-\ell:T}).$$

Doing this yields a useful by-product. Maximizing the ELBO also drives the KL divergence between $q_\phi(\boldsymbol{z}_{T-\ell:t}|\boldsymbol{x}_{T-\ell:T})$ and $p_\theta(\boldsymbol{z}_{T-\ell:t}|\boldsymbol{x}_{T-\ell:t})$ to be small. We will exploit this fact in our experiments to enable cutoff prediction. In particular we will directly approximate Equation 4 by:

$$p(\boldsymbol{x}_{T+1}|\boldsymbol{x}_{T-\ell:T}) \approx \int p_\theta(\boldsymbol{x}_{T+1}|\boldsymbol{z}_{T+1}) p_\theta(\boldsymbol{z}_{T+1}|\boldsymbol{z}_T) q_\phi(\boldsymbol{z}_{T-\ell:T}|\boldsymbol{x}_{T-\ell:T}) d\boldsymbol{z}_{T-\ell:T+1}$$

$$\approx \frac{1}{K} \sum_{k=1}^{K} p_\theta(\boldsymbol{x}_{T+1}|\boldsymbol{z}_{T+1}) p_\theta(\boldsymbol{z}_{T+1}|\boldsymbol{z}_T^{(k)}) \tag{5}$$

with $\boldsymbol{z}_T^{(k)}$ being the last-time-step marginal of the $k$th of $K$ samples from $q_\phi(\boldsymbol{z}_{T-\ell:T}|\boldsymbol{x}_{T-\ell:T})$. The predictive runtimes given by this technique can now be used to determine the throughput-maximizing cutoff in the objective given by Equation 2.

### 3.1.4 Handling Censored Run-times

As described, we use the learned inference network to predict future cutoffs rather than the generative model. Because variational inference jointly learns the model parameters along with the inference network, we could theoretically use an inference algorithm such as SMC (Doucet et al., 2001) for more accurate estimates of the true posterior predictive distribution. However, our cutoff prediction must be done in an amortized setting because we rely on it to be set for a gradient run prior to the updates returning from the workers. In a setting requiring fast, repeated inference, using an amortized method is often the only feasible approach, especially in large complex models.

However, when using amortized inference, there is a practical issue of dealing with partially observed, censored data. Since at run-time we are only waiting for $c$ gradients up to the cutoff, and are in fact actually killing the straggling workers, we do not have the run-time information from the straggling workers that would have finished past the cutoff. Inference in the generative model could directly be made able to deal with censored data, however our inference network runs an RNN which was trained on fully observed run-time vectors and therefore requires fully observed input to function correctly. Because of this, we deploy an effective approximate technique for imputing the missing worker runtime values, which samples a new uncensored data point for every worker whose gradients are dropped. Because we push estimates of the approximate posterior through the generative model, we have a predictive run-time distribution for the *current* iteration of SGD before receiving actual updates from any worker. When eventually the cutoff is reached, and the corresponding rate censor is observed, we are left with run-time distributions truncated at $\tilde{x}_{(c)}$:

$$p(\tilde{x}; \tilde{x} > \tilde{x}_c) = \frac{p(\tilde{x})}{\int_{\tilde{x}_{(c)}}^{\infty} p(\tilde{x})d\tilde{x}} \qquad (6)$$

where we have left off the time index for clarity and $\tilde{x}$ is any one of the censored worker runtime observations. When a censored value is required, we take its corresponding predicted run-time distribution and sample from its right tailed truncation to get an approximate value for that missing run-time. We find that this method works well to propagate the model forward, leading to still accurate predictions.

## 4 Experiments

To test our model's ability to accurately predict joint worker runtimes, we perform experiments by training 4 different neural network architectures on one of two clusters of different architectures and sizes. To train the BDSGD generative models used, we first train the neural network architecture of interest using fully synchronous SGD and use the recorded worker runtimes during each SGD iteration to learn a corresponding generative model of that particular neural network's gradient computation times on the cluster. As we will highlight, BDSGD model-based estimates of expected runtimes are sufficient to derive a straggler cutoff from their order statistics that leads to increased throughput and/or faster training times in real world situations.

### 4.1 Small Compute Cluster

On one cluster comprised of four nodes of 40 logical Intel Xeon processors, we benchmark Elfving and BDSGD cutoff predictors against the fully synchronous and fully asynchronous SGD with a 158-worker model by using each method to train a 2-layer CNN on MNIST classification. At this scale, and on a small neural network model, we are still able to deploy a Hogwild training scheme that does not diverge.

This cluster uses a job scheduler that allows jobs to be run concurrently on each node. From one of the fully synchronous SGD runs used to gather runtime data for BDSGD model learning, we find that 40 of the workers localized to one machine node experience a temporary $2\times$ slowdown in gradient computation times, which we believe to have been caused by another job batched onto the node. Figure 2 shows the transitional window of SGD worker runtimes, where the first 75 SGD iterations experience this slowdown before returning to normal for the remaining 125 iterations. In the case of 25% slow workers running $2\times$ slower, naive synchronous SGD decreases throughput by 50%. BDSGD, however, is able to correctly ignore all 40 slow workers (Fig. 3a), leading to near-optimal throughput despite the $2\times$ slowdown in SGD iteration time.

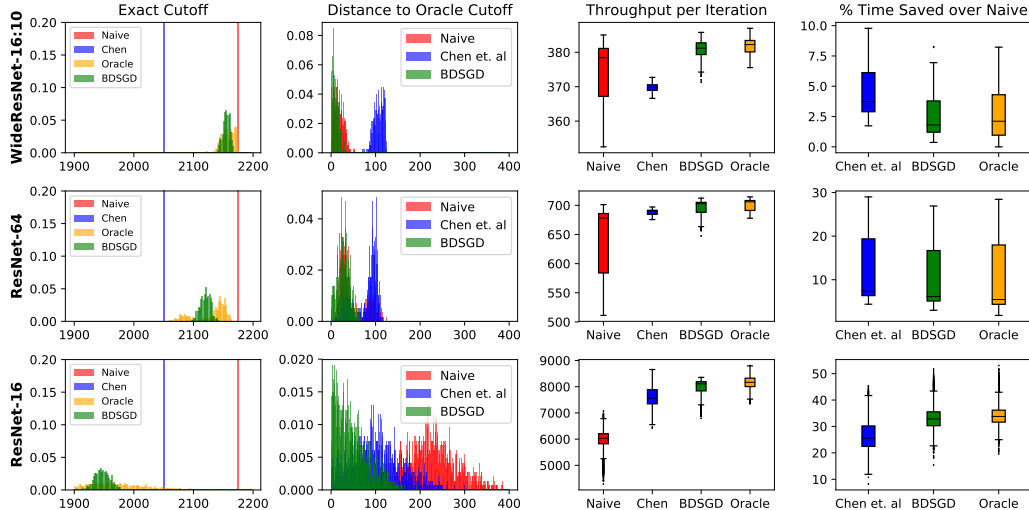

Figure 4: Comparison of BDSGD against Chen et al. on synchronous test data for three neural network training runs with no early worker termination. From left: histogram of cutoff set by each method (Chen et al. always uses 2051 workers and naive always uses all 2175 workers), histogram of the absolute difference between chosen cutoff and the oracle cutoff which would achieve highest throughput, quartile-boxplot of gradient computation throughput per iteration, and percent wall clock time saved using each method over fully synchronous (naive) method. We observe that BDSGD best predicts the oracle cutoff, which leads to highest throughput on all three cases, and highest expected wall-clock savings when BDSGD's average cutoff is lower than that fixed by Chen et al. (ResNet-16 model).

Figure 3b shows that our method achieves the fastest convergence to the lowest loss among comparison methods performing synchronous SGD. Hogwild outperforms our approach in wall-clock time, but its convergence is to a higher validation loss, as seen in the tails of the loss curves. Although not shown in subsequent experiments, Hogwild training diverges on larger clusters.

## 4.2   Large Scale Computing

On a large compute cluster, we use 32 68-core CPU nodes of a Cray XC40 supercomputer to compare 2175-worker BDSGD runs against the Chen et al. cutoff and naive methods on training three neural network architectures for CIFAR10 classification: a WideResNet model (Zagoruyko and Komodakis, 2016) and 16 and 64 layer ResNets. Using increasingly larger networks and batchsizes allows us to benchmark our speedup in situations called for by the large amount of recent work on training with 10K+ mini-batch sizes and high learning rates, (e.g. : Codreanu et al. (2017); You et al. (2017a,b); Smith et al. (2017)). For generative models of these neural network models on this cluster, we empirically find that training the latent variable model to directly emit sorted runtime order statistics is both faster to train and more accurate. Sampled draws from these distributions are reordered as before to calculate the predicted maximum throughput.

Unlike the 158-worker cluster, jobs on the Cray XC40 are sequestered to dedicated nodes by the scheduler. In Figure 4, we compare BDSGD and Chen et al.'s fixed cutoff method on validation sets in order to isolate the effect of accurate cutoff prediction on expected iteration throughput and speedup. BDSGD provides the best model of these runtimes, subsequently leading to near optimal throughput.

Figure 5 shows the wall-clock training times and throughputs achieved under real workloads for each training method on the same three neural networks. Comparing the production throughput in rows 1 and 2 of Figure 5 to the expected throughput in rows 1 and 2 of Figure 4, all training methods experience a small drop-off in throughput when run in production due to communication costs and other additional overhead. For these two neural networks, BDSGD is still shown to produce the highest throughput when used during training. For training the WideResNet, Chen et al.'s method achieves a 1.2% speedup in wall-clock training time whereas BDSGD is able to calculate 5% more

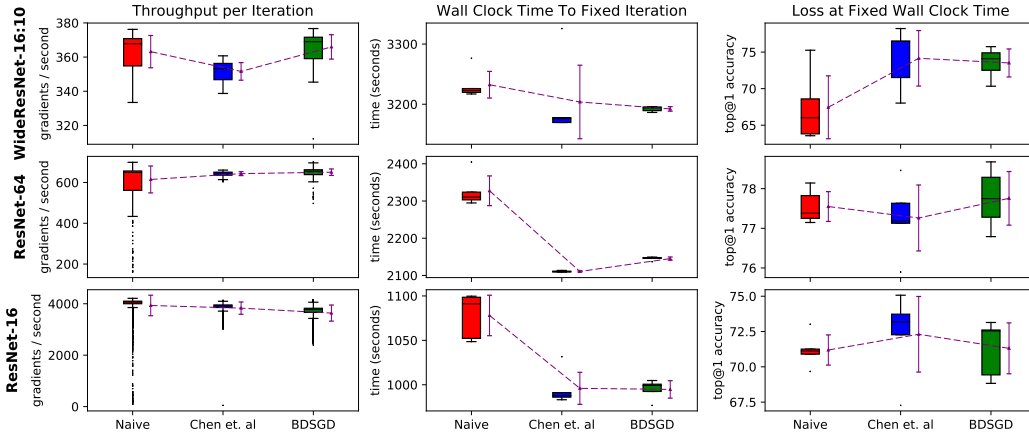

Figure 5: Production training runs comparing BDSGD, Chen et al., and fully synchronous (naive) methods. The neural networks on each row are trained to convergence using the three training methods, five times each. Each row in this figure corresponds to the same row in Figure 4. The columns show quartile-boxplots and mean±std (purple error bar). From left to right: iteration throughput achieved during training, wall-clock time to fixed iteration (400, 600, and 1500 for WideResNet, ResNet-64, and ResNet-16, respectively), and validation loss at a fixed wall-clock time (set to the wall-clock time at 50% of the total training time taken by Chen et al.'s method). The two big neural network models, ResNet-64 and WideResNet-16:10, achieve the highest throughput when using BDSGD, training to roughly equal or better validation loss at a fixed wall-clock time, while improving total training time by roughly the same as Chen et al. despite setting much higher cutoffs. The ResNet-16 model demonstrates the ability to run a modified BDSGD on a small network, where the amortized inference exceeds gradient computation time on average.

gradients to achieve a 1% speedup in wall-clock training time. Similarly, BDSGD achieves a 7.8% speedup in wall-clock training time while using 3.2% more gradients than Chen et al.'s method, which achieves a 9.3% speedup.

In the final row of Figure 5, we demonstrate the ability for BDSGD prediction to be robust to the scenario in which performing amortized inference in the generative model exceeds the time it takes for workers to finish their gradient computations. Here, we use a modified variant of DBSGD that fixes the predicted cutoff for ten iterations at a time in order to avoid being bottlenecked at every iteration by the parameter server cutoff prediction. In doing so, we show in the final row of Figure 5 that one may still achieve a 7.6% speedup with sparse predictions from the generative model.

All training methods for the three neural network models in Figure 5 train to a similar final held-out validation accuracy.

# 5 Discussion

We have presented a principled approach to achieving higher throughput in synchronous distributed gradient descent. Our primary contributions include describing how a model of worker runtimes can be used to predict order statistics that allow for a near optimal choice of straggler cutoff that maximizes gradient computation throughput.

While the focus throughout has been on on vanilla SGD, it should be clear that our method and algorithm can be nearly trivially extended to most optimizers of choice so long as they are stochastic in their operation on the training set. Most methods for learning deep neural network models today fit this description, including for instance the Adam optimizer (Kingma and Ba, 2014).

We conclude with a note that our method implicitly assumes that every mini-batch is of the same computational cost in expectation, which may not always be the case. Future work could be to extend the inference network further (Rezende and Mohamed, 2015) or to investigate variable length input in distributed training as in Ergen and Kozat (2017).

**Acknowledgments**

Michael Teng is supported under DARPA D3M, under Cooperative Agreement FA8750-17-2-0093 and partially supported by the NERSC Big Data Center under Intel BDC. This research used resources of the National Energy Research Scientific Computing Center (NERSC), a DOE Office of Science User Facility supported by the Office of Science of the U.S. Department of Energy under Contract No. DE-AC02-05CH11231. We acknowledge Intel for their funding support and we thank members of the UBC PLAI group for helpful discussions.

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
