[Supplementary Material]

# 1 Supplementary Material

For a complete treatment of the methods used, we detail the generative network model $p_\theta$ and the inference network $q_\phi$ designs. We also provide pseudocode for the parameter server and worker client algorithms. Finally, we specify experimental training and testing parameters.

## 1.1 BDSGD generative model

The generative model factorises as latent variable state space model:

$$p_\theta(\boldsymbol{x}_{T-\ell:T}, \boldsymbol{z}_{T-\ell:T}) = \prod_{i=T-\ell}^{T} p_\theta(\boldsymbol{z}_i|\boldsymbol{z}_{i-1}) \prod_{i=T-\ell}^{T} p_\theta(\boldsymbol{x}_i|\boldsymbol{z}_i)$$

where $p_\theta(\boldsymbol{z}_t|\boldsymbol{z}_{t-1})$ is parametrized as:

$$\boldsymbol{z}_t \sim \mathcal{N}(G_\theta(\boldsymbol{z}_{t-1}), H_\theta(\boldsymbol{z}_{t-1}))$$
$$G_\theta(\boldsymbol{z}_{t-1}) = (1 - g_t) \cdot \text{1-MLP}(\boldsymbol{z}_{t-1}, \text{Identity}) + g_t \cdot \text{2-MLP}(\boldsymbol{z}_{t-1}, \text{ReLU}, \text{Identity})$$
$$g_t = \text{2-MLP}(\boldsymbol{z}_{t-1}, \text{ReLU}, \text{Sigmoid})$$
$$H_\theta(\boldsymbol{z}_{t-1}) = \text{1-MLP}(\text{ReLU}(G_\theta(\boldsymbol{z}_{t-1})), \text{Softplus})$$

and $p_\theta(\boldsymbol{x}_t|\boldsymbol{z}_t)$ is parametrized as:

$$\boldsymbol{x}_t \sim \mathcal{N}(I_\theta(\boldsymbol{z}_t), J_\theta(\boldsymbol{z}_t))$$
$$I_\theta(\boldsymbol{z}_t) = \text{2-MLP}(\boldsymbol{z}_{t-1}, \text{Identity}, \text{Identity})$$
$$J_\theta(\boldsymbol{z}_t) = \text{2-MLP}(I_\theta(\boldsymbol{z}_t), \text{ReLU}, \text{Softplus})$$

where $n\text{-MLP}(\,\cdot\,, A, B, ...)$ denotes an $n$-layer multilayer perceptron with nonlinearities, $A$, $B$, etc.

## 1.2 BDSGD inference network

The inference network is a structured left-right model, conditioning on observations from the future and past:

$$q_\phi(\boldsymbol{z}_{T-\ell:T}|\boldsymbol{x}_{T-\ell:T}) = \prod_{i=T-\ell}^{T} q_\phi(\boldsymbol{z}_i|\boldsymbol{z}_{T-\ell:i}, \boldsymbol{x}_{T-\ell:T})$$

where $*$ denotes scalar multiplication below and $q_\phi(\boldsymbol{z}_i|\boldsymbol{z}_{T-\ell:i}, \boldsymbol{x}_{T-\ell:T})$ is parametrized as:

$$\boldsymbol{z}_t \sim \mathcal{N}(K_\phi(\boldsymbol{z}_{t-1}, \boldsymbol{x}_{T-\ell:T}), L_\phi(\boldsymbol{z}_{t-1}, \boldsymbol{x}_{T-\ell:T}))$$
$$K_\phi(\boldsymbol{z}_{t-1}, \boldsymbol{x}_{T-\ell:T}) = \text{1-MLP}(h_{\text{out}}, \text{Identity})$$
$$L_\phi(\boldsymbol{z}_{t-1}, \boldsymbol{x}_{T-\ell:T}) = \text{1-MLP}(K_\phi(\boldsymbol{z}_{t-1}, \boldsymbol{x}_{T-\ell:T}), \text{Softplus})$$
$$h_{\text{out}} = \frac{1}{3} * (\text{1-MLP}(\boldsymbol{z}_{t-1}, \text{Tanh}) + h_{\text{left}} + h_{\text{right}})$$
$$h_{\text{left}} = \text{RNN}(\boldsymbol{x}_{T-\ell:t}, \text{ReLU})$$
$$h_{\text{right}} = \text{RNN}(\boldsymbol{x}_{t+1:T}, \text{ReLU})$$

## 1.3 Parameter server and worker design

We detail the parameter server and worker client pseudo code when using BDSGD.

| **Algorithm 1** parameter server | **Algorithm 2** worker |
|---|---|
| 1: **Input:** model $p_\theta$, inference network $q_\phi$, | 1: **Input:** data split $X$, |
| 2:      learning rate $\alpha$, worker count $N$ | 2:      local batch size $m$, |
| 3: | 3:      worker index $n$ |
| 4: runtimes $\leftarrow \{\}$ | 4: |
| 5: $t \leftarrow 0$ | 5: **procedure** compute_gradient($\theta$): |
| 6: **while not** converged **do** | 6:      $x \leftarrow$ timestamp() |
| 7:      $G \leftarrow \langle\rangle$ | 7:      **for** $i = 1$ **to** $b$ |
| 8:      $\boldsymbol{x}_t \leftarrow \langle\rangle$ | 8:        $x^{(z)}, y^{(z)} \leftarrow$ sample($X$) |
| 9:      $c, \tilde{\boldsymbol{x}}_{(1):(N)} \leftarrow$ predict($p_\theta, q_\phi$, runtimes) | 9:        $\langle\nabla_\theta F\rangle^n \leftarrow \frac{1}{m}\nabla F(\theta, x^{(z)}, y^{(z)})$ |
| 10: | 10:      **end for** |
| 11:      **for** $0 < j \leq c$ **do** | 11:      $x^n \leftarrow$ timestamp() $- x$ |
| 12:        **receive** $\langle\nabla_\theta F\rangle^n, x^n \leftarrow$ any worker | 12:      **send** $\langle\nabla_\theta F\rangle^n, x^n \rightarrow$ parameter server |
| 13:        $\boldsymbol{x}_t \leftarrow \boldsymbol{x}_t \cup x^n$ | 13: **end procedure** |
| 14:        $G \leftarrow G + \langle\nabla_\theta F\rangle^n$ | 14: |
| 15:      **end for** | 15: $\theta \leftarrow$ initialize($\theta$) |
| 16:      $\theta' \leftarrow \theta - \alpha\frac{1}{c}G$ | 16: **while not** converged **do** |
| 17:      **send** $\theta' \rightarrow$ all workers | 17:      **spawn** thread **run** compute_gradient($\theta$) |
| 18: | 18:      **await receive** $\theta' \leftarrow$ parameter server |
| 19:      $x_c \leftarrow \max(\boldsymbol{x}_t)$ | 19:      $\theta \leftarrow \theta'$ |
| 20:      **for** $c < k \leq N$ **do** | 20:      **terminate** thread |
| 21:        $x^k \leftarrow$ sample $p(\tilde{x}_{(k)}; \tilde{x}_{(k)} > x_c)$ | 21: **end while** |
| 22:        $\boldsymbol{x}_t \leftarrow \boldsymbol{x}_t \cup x^k$ | |
| 23:      **end for** | |
| 24:      runtimes$\{\text{key}: t\} \leftarrow \boldsymbol{x}_t$ | |
| 25:      **increment** t | |
| 26: **end while** | |

The await parameter server udpate and thread termination mechanism in lines 18-20 of Alg. 2 can be implemented using threads, async promises, or continuous polling of parameter server socket.

## 1.4 Experimental runs

We train 4 distinct neural network models in our experiments: 2-layer CNN on MNIST classification and ResNet-16, ResNet-64, and WideResNet-16x10 on CIFAR10 classification.

The CNN is trained with a batch size of 10112 and fixed learning rates of 0.64 for BDSGD, fully synchronous SGD, and Elfving SGD runs on a 158-worker cluster. The CNN is also trained with Hogwild and a learning rate of 0.004 (scaled down from 0.64 by the number of workers).

The remaining WideResNet and ResNets are trained on a 2175-worker cluster uses SGD with 0.9 momentum and a linearly scaled learning rate warmup for 50 iterations from an initial learning rate of 0.001. After the warmup phase, the WideResNet model uses a learning rate of 0.12 decayed by 80% at 200 iterations and 50% more at 350 iterations, ResNet-64 uses a 0.18 learning rate decayed by 80% at 250 iterations and 80% more at 450 iterations, and ResNEt-16 uses a 0.06 learning rate with no decay. The models are trained to 400 iterations, 600 iterations, and 1500 iterations, respectively.

Each of these 4 models is trained with fully observed runtime information from naive synchronous training runs of the neural network on the corresponding cluster. All generative models use a 50-dimensional latent variable with 30-dimensional and 50-dimensional MLP link functions for the transfer and emission functions, respectively. The RNN in the proposal network uses 200 hidden units.

For BDSGD on CNN training, we train the generative model with Adam, using a 0.00004 learning rate, 0.96:0.999 beta1 and beta2 values, gradient clipping at 20.0, weight decay of 0.6, and a learning rate decay of 0.99996. and a minibatch size of 8. For the 3 BDSGD models on the 2175-cluster, we use Adam with a minibatch size of 20 and a 0.00002 learning rate. All models are trained with a twenty-timestep lag, i.e. $\ell = 20$.