[Reviews · NeurIPS 2018]

Reviewer 1



Summary: This paper presents a new algorithm called Bayesian Distributed SGD to mitigate the straggler problem when training deep learning on parallel clusters. Unlike Synchronous Distributed SGD approach where a fixed cut-off (number of workers) is predefined, BDSGD uses amortized inference to predict workers’ run-times and derive a straggler cut-off accordingly. By discarding sub-gradients from slow workers, the overall system throughput increases and hence faster time to convergence. BDSGD models the joint run-time behaviour of workers which are likely to be correlated due to the underlying cluster architecture. The approach is incorporated as part of the parameter server framework, deciding which sub-gradients to drop in each iteration. Strength: The proposed idea of adaptive cut-off and predicting joint worker runtime through amortized inference with variational auto encoder loss is novel and very interesting. Well-written and straightforward for readers. The evaluations are done with real-world datasets and different workloads. Both model convergence and system profile are reported. Weakness and comments: To handle censored observations, the authors propose to sample uncensored data points to impute the missing run-times. Will such sampling suffer from bias given that the uncensored data points are only available when the workers successfully pass the cut-off: uncensored run-times tend to be shorter than censored ones? In this case, wouldn’t this imputation negatively affect the subsequence inference performance? Since a new cut-off is derived in every iteration, can the authors also demonstrate more detail on the cut-off value over time? The training dataset is generated by instrumenting the cluster once and then used to train the model to encode worker’s runtime behaviour. If the cluster utilisation at test-time is very different from that when training dataset is gathered, will the model still be as accurate as expected? Also, since gradients produced by stragglers are dropped due to the cut-off mechanism, it seems difficult to collect the actual straggler run-times to further adapt the model (plus the effect of the imputation). In figure 3, would be useful to include convergence based on Chen et al technique. Can the authors comment why BDSGD results in high-variance throughput while Chen et al technique has a relatively stable trend? In the context of high communication cost, why is BDSGD’s throughput significantly lower than Chen et al’s? Can the authors give explanation on this? Can BDSGD be used with distributed parameter servers? *UPDATE* I would like to thank the authors for the rebuttal. I have read the discussion provided and am happy to increase the overall score.

Reviewer 2



========================================Update after authors' rebuttal======================================== I am generally happy with the authors' rebuttal, but would like to add a response to the reply, "For our purposes, we find that batch-size and objective function gain to be correlated in the literature, assuming strong convexity and Lipschitz-continuity ... ". These are rather strong assumptions that are not necessarily true, especially for deep neural networks. Furthermore, finding this optimal m (after taking the product with time-to-m) is non-linearly dependent on the local convexity and Lipschitzness, i.e. you really should be optimizing for objective function gain! To be clear, the paper remains interesting irrespective of this, but I do think it would be a good direction for future research to properly understand the difference between optimizing throughput versus optimizing objective function gain. ================================================================================ This paper describes a method to improve on Chen et al, by automatically learning and deciding on the number of workers to wait for in distributed synchronous SGD, so as maximize "throughput" and avoid the straggler effect. This is done by using deep linear dynamical models to estimate worker run-times. The number of workers to wait for is then dynamically determined via model inference and maximizing the expected throughput at the next iteration. Experiments conducted show that the method can provide improvements over the static cutoff of Chen et al, and over a baseline dynamic Elfving cutoff. This work improves over Chen et al by introducing a dynamic cutoff in a principled manner, further improving the state-of-the-art for synchronous SGD. In spite of the seemingly heavy machinery, the experiments show that a dynamic estimation works better than the static cut-off. This work could generate discussion in the systems-ML community, further the debate between asynchronous and synchronous distributed methods. I have a number of questions / suggestions: - The technique targets maximization of "throughput", which does not take into account the improvement in the objective function. What we're really interested in is the "rate of objective improvement", or roughly speaking, \Delta(c) / \tilde{x}_{(c)}, where \Delta(c) is the decrease in the objective function using gradients from c workers. Despite the claims (line 112-113) that optimizing throughput "handles tradeoff between iteration speedup and learning signal reduction", I do not see how it addresses the optimization problem / gradient variance. To make this more concrete, consider the optimization of a convex problem. At the start, we would expect all stochastic gradients to be generally in agreement, in which case, it might be best to use few workers, since \Delta may not change much with larger c. As we approach the opt, stochastic noise dominates, and \Delta increases quickly with larger c, so we would want to use as many workers as possible while still mitigating for stragglers. - In the experiments, comparison between methods is often made in terms of time to reach X iterations. However, the real measure should be time to reach a loss or accuracy target. (After all, the entire point of cutoff methods is to trade-off between the statistical efficiency of synchronous methods and the throughput of asynchronous methods to achieve the best wall-clock time to accuracy, and not the best wall-clock time to target number of iterations.) - As mentioned above, a lot of work goes into estimating the dynamic cutoff point. How much computation time does this take relative to the gradient computation? - How does the cutoff point change over the course of the optimization? If it stabilises quickly, can we switch to a static method, or at least reduce the frequency at which the cutoff is changed? - How would the proposed method compare with a naive approach of constructing an empirical CDF (a la Figure 2) and using that to determine cutoff?

Reviewer 3



There is substantial interest at present in scaling up parallel training of DNNs across multiple machines. While much of this work has been focused on asynchronous SGD, it has recently been shown that one can increase the batch size of synchronous training to tens of thousands (on ImageNet) with only a small drop in the final test accuracy. However parallel synchronous training suffers from the straggler effect, whereby a small tail of machines may take substantially longer to compute gradients than the bulk, potentially significantly reducing the training speed. This paper proposes to use amortised inference to learn a generative model which predicts when to discard slow gradients. Originality: The paper builds on a recent work [1] which proposed to simply discard a fixed fraction of slow gradients each update. They claim to be the first to learn a model to optimise the decision process during training. Significance: This seems a sensible and valuable contribution, with promising early results. Quality: The quality seemed good, although I am not familiar with this field. Some specific comments: 1. I’d recommend that the authors couple the batch size, which varies from iteration to iteration, to a linear scaling relationship with the learning rate. If one assumes that slow workers are not correlated with particular training examples, this will ensure that training is not adversely affected by fluctuations in the batch size. 2. It would have been nice to see some back of the envelope estimates of how much wall-clock time could in principle be saved. Ie how long on average do machines sit idle in a standard run, and how much is this reduced by the simple baseline of [1]? 3. I’m confused by section 3.1.1: The authors state they assume the run times are drawn from a Gaussian distribution, but surely this cannot be since all run-times must be positive? (To make such an assumption valid the mean would have to significantly exceed the standard deviation, in which case there is no straggler effect) 4. Would it be worth letting a small fraction of the stragglers run to completion to gather information on the tail of the run-time distribution? 5. The model strikes me as rather complicated, especially since the end task is to infer a single number between 0 and 1 (the fraction of gradients to collect). Likely in most cases the distribution does not change significantly during training. Would a simple bandit or regression model not achieve most of the gains reported here? Update: I thank the authors for their helpful response. I note that the beta distribution provides a natural probability distribution over [0,\infty], however I am happy with the argument that this is best approximated by a Gaussian for simplicity, and remain positive about the work.